# Integrating Bayesian and Discriminative Sparse Kernel Machines for Multi-class Active Learning

**Weishi Shi**
Rochester Institute of Technology
ws7586@rit.edu

**Qi Yu**
Rochester Institute of Technology
qi.yu@rit.edu

## Abstract

We propose a novel active learning (AL) model that integrates Bayesian and discriminative kernel machines for fast and accurate multi-class data sampling. By joining a sparse Bayesian model and a maximum margin machine under a unified kernel machine committee (KMC), the proposed model is able to identify a small number of data samples that best represent the overall data space while accurately capturing the decision boundaries. The integration is conducted using the maximum entropy discrimination framework, resulting in a joint objective function that contains generalized entropy as a regularizer. Such a property allows the proposed AL model to choose data samples that more effectively handle non-separable classification problems. Parameter learning is achieved through a principled optimization framework that leverages convex duality and sparse structure of KMC to efficiently optimize the joint objective function. Key model parameters are used to design a novel sampling function to choose data samples that can simultaneously improve multiple decision boundaries, making it an effective sampler for problems with a large number of classes. Experiments conducted over both synthetic and real data and comparison with competitive AL methods demonstrate the effectiveness of the proposed model.

## 1   Introduction

While more labeled data tends to improve the performance of supervised learning, labeling a large number of data samples is labor intensive and time consuming. Furthermore, obtaining accurate labels may be highly challenging for many specialized domains, such as medicine and biology, where expert knowledge is required for understanding and extracting the underlying semantics of data. Active Learning (AL) provides a promising direction to use a small subset of labeled data samples to train high-quality supervised learning models in a cost-effective way. Consequently, AL has been successfully applied to various applications [1, 2, 3].

A large number of AL models have been developed for different types of supervised learning models. However, the design of the data sampling strategy is usually limited by the learning models, which are not designed specifically for AL purpose. For example, max-margin based classifiers, such as support vector machines (SVMs), are widely used for sampling purpose in AL. However, as they are essentially designed for the classification task, using them directly for sampling might lead to a slow convergence. Figure 1a illustrates such behavior in existing models. Assume that the two middle clusters contain 80% of data samples. Hence, it is highly likely that the initially labeled samples are from these two clusters, which give the initial decision boundary as shown by the dashed line. Then, samples in the middle clusters will continue to be sampled as they are close to the current decision boundary. This will cause a very slow convergence to the true decision boundary shown as the solid line. Furthermore, since the model performance over iterations stays roughly the same, it may cause AL to terminate. This is undesirable, because the true decision boundary is never discovered. Such behavior is intrinsic to the classifier, which primarily focuses on exploiting the current decision boundary rather than exploring the entire data distribution for more effective data sampling.

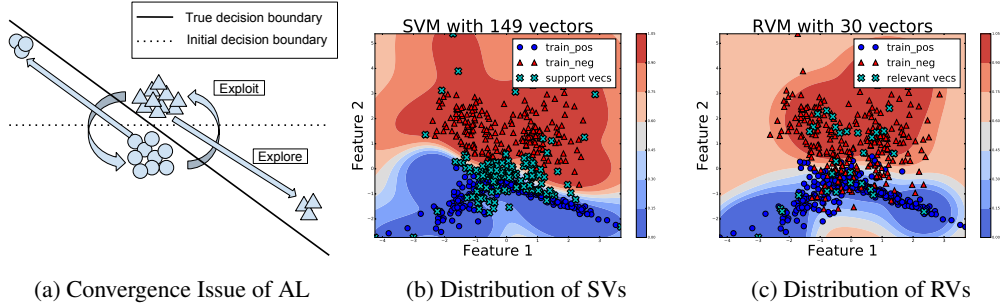

| (a) Convergence Issue of AL | (b) Distribution of SVs | (c) Distribution of RVs |

Figure 1: (a) Undesired convergence behavior of AL; (b) Distribution of SVs; (c) Distribution of RVs

To address the undesired convergence behavior of existing AL models, we propose a novel a kernel machine committee (KMC) based model that integrates Bayesian and discriminative sparse kernel machines for multi-class active learning. The KMC model naturally extends the sampling space from around the current decision boundaries to other critical areas through the representative data samples identified by the sparse Bayesian model. More specifically, the proposed KMC sampler incorporates a relevant vector machine (RVM), which is a Bayesian sparse kernel technique, to identify data samples (referred as relevance vectors, or RVs) that capture the overall data distribution. By further augmenting the SVM with a RVM, the KMC model is able to choose data samples that provide a good coverage of the entire data space (by maximizing the data likelihood) while giving special attention to the critical areas for accurate classification (by maximizing the margins of decision boundaries). Figures 1b and 1c demonstrate the complementary distribution of RVs and SVs (support vectors) and how they cover different critical areas in the data space, where most SVs are located near to the decision boundary while most RVs are in the densely distributed areas of the two classes. There are also much less RVs than SVs, implying that RVM is even a sparser model than SVM [4]. The sparse nature of both RVM and SVM makes their combination an ideal choice for AL.

In essence, the KMC joins a sparse Bayesian model (RVM) with a maximum margin machine (SVM) to choose data samples that meet two key properties simultaneously: (1) providing a good fit of the overall data distribution, and (2) accurately capturing the decision boundaries. We propose to use the maximum entropy discrimination (MED) framework [5, 6] to seamlessly integrate these two distinctive properties into one joint objective function to train the KMC for multi-class data sampling in AL. Furthermore, the objective function can be equivalently expressed as combination of a likelihood term with the generalized entropy [7]. This deeper connection implies that the KMC model is able to choose data samples that are most instrumental to tackle the difficult non-separable classification problems (as shown in our experiments). In contrast, the SVM based models need a much large number of SVs (hence more labeled data) to accurately capture the more complex decision boundaries. Our main contribution is threefold: (i) a novel kernel machine committee (KMC) model that seamlessly unifies Bayesian and discriminative sparse kernel machines for effective data sampling; (ii) a principled optimization framework by leveraging convex duality and sparse structure of KMC to efficiently optimize the joint objective function; and (iii) a novel sampling function that combines key model parameters to choose data samples that can simultaneously improve multiple decision boundaries, making it an effective sampler for problems with a large number of classes.

## 2 Related Work

Uncertainty sampling is one of the most commonly used sampling method for AL, where the informativeness of the unlabeled data point is determined by its distance to the decision boundaries [8, 9]. In order to better accommodate multi-class problems, Culotta and McCallum [10] propose a sampling method based on the predictive probability over the sample pool where the data point with the smallest probability of its predicted class is sampled. However, this method prefers trivial data samples with evenly distributed predictive probabilities. To address this, a Best-versus-Second Best (BvSB) model is proposed to choose the data sample whose probabilities of the most and second most classes are closest to each other [10]. But BvSB only focuses on the two most probable classes while the probability distribution of other classes is ignored. As a result, this method is less effective with more classes. Entropy-based methods obtain a complete view of uncertainty over all classes to conduct effective AL sampling. But the lack of training samples in the beginning of the AL impedes the accurate estimation of entropy. In fact, all sampling methods that rely on the probability output of

SVM should be dealt with caution as the probabilities are estimated by fitting an additional logistic regression model over SVM scores. Thus, the estimated probability might not reflect the true behavior of the SVM as a discriminative model [11].

Compared to discriminative models (e.g., SVMs), generative models can be more naturally used for multi-class AL. Roy et al. propose a sampling method based on Naive Bayes using the expectation of future classification error as the sampling criterion [12]. Kottke et al. propose a multi-class probabilistic AL model (McPAL) according to the expectation of the classification error from clusters of unlabeled data [13]. Both methods are computationally intensive, making them hard to be applied to real-time AL. Furthermore, since the learning objective (e.g. maximum likelihood) of generative models are not specifically designed for discrimination [14], the model performance might be less competitive when AL is complete as compared to the discriminative models.

There are also existing models that utilize the properties of the data space or the trained model for effective data sampling. A convex-hull based AL model is developed to avoid sampling less informative data points that are close to the current support vectors [15]. A similar strategy is adopted in [16], where data samples with furthest distance to its closest relevance vectors are sampled. The QUIRE model combines the clustering structure of unlabeled data with the class assignments of the labeled data, allowing it to choose samples both informative and representative [17]. However, this model is designed for binary problems. Different from all existing works, the proposed KMC model leverages the complementary behavior of Bayesian and discriminative sparse kernel machines and systematic integrates them for effective data sampling for multi-class AL.

## 3    Kernel Machine Committee based Active Learning

Let $\mathbf{X} = \{\mathbf{x}_1, ..., \mathbf{x}_M\}$ denote a training set with $M$ data samples and $\mathbf{y} = \{y_1, ..., y_M\}$ be their corresponding labels. Let's consider the binary-class case where $\forall y_i \in \mathbf{y}, y_i \in \{-1, +1\}$ and the multi-class problems can be achieved via the one-versus-the-rest strategy. The conditional distribution of label $y_i$ is given by $p(y_i = 1|\mathbf{w}, \mathbf{x}_i) = \sigma(\mathbf{w}^T \phi(\mathbf{x}_i))$, where $\sigma$ is the logistic sigmoid function, $\mathbf{w}$ is the coefficient, and $\phi(\mathbf{x}_i)$ is feature vector of $\mathbf{x}_i$. For RVM, we set $\phi_j(\mathbf{x}_i) = k(\mathbf{x}_i, \mathbf{x}_j)$ with $k(\cdot, \cdot)$ being a kernel function. We further place a prior over the coefficient $\mathbf{w}$ with hyperparameter $\boldsymbol{\alpha} = (\alpha_1, ..., \alpha_M)$, given by $p(\mathbf{w}|\boldsymbol{\alpha}) = \prod_j \mathcal{N}(w_j|0, \alpha_j^{-1})$. Having a separate hyperparameter $\alpha_i$ for each coefficient $w_i$ will ensure model sparsity through automatic relevance determination (ARD) [4]. In particular, during the parameter learning process, some of the $\alpha_i$ will be driven to infinity, which has the effect of making the corresponding $w_i$ approach zero. As a result, the associated data sample $\mathbf{x}_i$ will be excluded from the RV set. The optimal $\boldsymbol{\alpha}$ can be determined through evidence approximation, which maximizes the log marginal likelihood of observed data given by $\ln p(\mathbf{y}|\mathbf{X}, \boldsymbol{\alpha}) = \ln \int \prod_i [p(y_i|\mathbf{x}_i, \mathbf{w})] p(\mathbf{w}|\boldsymbol{\alpha}) d\mathbf{w}$.

Since the likelihood term $p(y_i|\mathbf{x}_i, \mathbf{w})$ is a logistic function, which is non-conjugate to the Gaussian prior $p(\mathbf{w}|\boldsymbol{\alpha})$, the integration can not be straightforwardly performed. By applying Jensen's inequality to the log function, we can obtain a lower bound of the log likelihood given by $\ln p(\mathbf{y}|\mathbf{X}, \boldsymbol{\alpha}) \geq \mathbb{E}_{q(\mathbf{w})}[\ln p(\mathbf{y}|\mathbf{X}, \mathbf{w})] - KL(q(\mathbf{w})||p(\mathbf{w}|\boldsymbol{\alpha}))$, where $q(\mathbf{w})$ is a variational distribution and the second term is the KL divergence between $q(\mathbf{w})$ and prior distribution $p(\mathbf{w}|\boldsymbol{\alpha})$. This change makes it possible to put parameter learning in RVM into the MED framework [5], which allows us to further integrate a set of margin-based constraints

$$\min_{q(\mathbf{w}), \boldsymbol{\alpha}, \boldsymbol{\xi}} \quad KL(q(\mathbf{w})||p(\mathbf{w}|\boldsymbol{\alpha})) - \mathbb{E}_{q(\mathbf{w})}[\ln p(\mathbf{y}|\mathbf{X}, \mathbf{w})] + C \sum_i \xi_i \qquad (1)$$

$$\text{subject to} \quad \forall i: \quad \mathbb{E}_{q(\mathbf{w})}[y_i f(\mathbf{w}, \mathbf{x}_i)] \geq -\xi_i, \quad \xi_i \geq 0, \quad \int q(\mathbf{w}) d\mathbf{w} = 1$$

where $\xi$'s are slack variables and $f(\mathbf{w}, \mathbf{x}_i)$ is a cost function, defined as $f(\mathbf{w}, \mathbf{x}_i) = \ln \frac{p(y_i=1|\mathbf{w}, \mathbf{x}_i)}{p(y_i=-1|\mathbf{w}, \mathbf{x}_i)} = \mathbf{w}^T \phi(\mathbf{x}_i), \forall i, y_i = -1; f(\mathbf{w}, \mathbf{x}_i) = \ln \frac{p(y_i=-1|\mathbf{w}, \mathbf{x}_i)}{p(y_i=1|\mathbf{w}, \mathbf{x}_i)} = -\mathbf{w}^T \phi(\mathbf{x}_i), \forall i, y_i = 1$, which ensures that a linear cost is introduced only for misclassified data samples.

Directly optimizing (1) is still challenging due to the likelihood term that follows a logistic function. We make further approximation by using an exponential quadratic function to lower bound the logistic function [18]: $\sigma(z) \geq \sigma(\gamma) \exp\{(z - \gamma)/2 - \lambda(\gamma)(z^2 - \gamma^2)\}$, where $\lambda(\gamma) = \frac{1}{2\gamma}(\sigma(\gamma) - \frac{1}{2})$. This approximation allows us to derive a lower bound of the likelihood term in (1).

**Lemma 1.** *For $\forall \boldsymbol{\gamma} \in R^M$, $\forall \mathbf{y} \in \{-1, +1\}^M$, there exists a lower bound of the likelihood of the logistic regression function that has an exponential quadratic functional form and satisfies:*

$$p(\mathbf{y}|\mathbf{w}, \mathbf{X}) \geq \prod_{i=1}^{M} \sigma(\gamma_i) \exp\left\{\frac{1}{2}(\mathbf{w}^T\phi(\mathbf{x}_i)y_i - \gamma_i) - \lambda(\gamma_i)([\mathbf{w}^T\phi(\mathbf{x}_i)]^2 - \gamma_i^2)\right\} \doteq h(\mathbf{w}, \boldsymbol{\gamma}) \quad (2)$$

where $\boldsymbol{\gamma} = (\gamma_1, ..., \gamma_m)^T$.

*Proof.* By leveraging the symmetry of the sigmoid function, we have

$$p(y_i = 1|\mathbf{w}, \mathbf{x}_i) = \sigma(\mathbf{w}^T\phi(\mathbf{x}_i)) \quad (3)$$

$$p(y_i = -1|\mathbf{w}, \mathbf{x}_i) = 1 - \sigma(\mathbf{w}^T\phi(\mathbf{x}_i)) = \sigma(-\mathbf{w}^T\phi(\mathbf{x}_i)) \quad (4)$$

Using $\sigma(z) \geq \sigma(\gamma)\exp\{(z-\gamma)/2 - \lambda(\gamma)(z^2 - \gamma^2)\}$, the conditional likelihood of $y_i$ is given by:

$$p(y_i|\mathbf{w}) = \sigma(y_i\mathbf{w}^T\phi(\mathbf{x}_i) \geq \sigma(\gamma_i)\exp\left\{\frac{1}{2}(\mathbf{w}^T\phi(\mathbf{x}_i)y_i - \gamma_i) - \lambda(\gamma_i)([y_i\mathbf{w}^T\phi(\mathbf{x}_i)]^2 - \gamma_i^2)\right\} \quad (5)$$

Substitute for $y_i^2 = 1$ and lemma 1 is proved. $\qquad\square$

Replacing the likelihood with $h(\mathbf{w}, \boldsymbol{\gamma})$ in (1), the final objective function of KMC is given by

$$\text{Objective (KMC):} \quad \min_{q(\mathbf{w}), \boldsymbol{\gamma}, \boldsymbol{\alpha}, \boldsymbol{\xi}} \quad KL(q(\mathbf{w})||p(\mathbf{w}|\boldsymbol{\alpha})) - \mathbb{E}_{q(\mathbf{w})}[\ln h(\mathbf{w}, \boldsymbol{\gamma})] + C\sum_i \xi_i \quad (6)$$

$$\text{subject to} \quad \forall i: \quad \mathbb{E}_{q(\mathbf{w})}[y_i f(\mathbf{w}, \mathbf{x}_i)] > -\xi_i, \quad \xi_i \geq 0, \quad \int q(\mathbf{w})d\mathbf{w} = 1$$

The first term is a regularizer of the variational distribution $q(\mathbf{w})$. The use of an ARD prior imposes the sparsity of $\mathbf{w}$, which guarantees the sparsity of the KMC. The second term approximates the negative log likelihood of the observed data and the last term brings in the maximum margin-based constraints. It is also worth to note that the expectation $\mathbb{E}_{q(\mathbf{w})}[f(\mathbf{w}, \mathbf{x}_i)]$ is taken over $q(\mathbf{w})$, which demonstrates the interplay of the Bayesian RVM model and the maximum margin SVM model. The integrated objective allows the KMC model to identify a small number of data samples, referred to as *KMC vectors*, which can describe the observed data well while accurately capturing the decision boundaries at the same time. In addition, by combining the first and third terms, they form a special case of generalized entropy [7]. Therefore, the KMC objective function can also be interpreted as minimizing the negative log likelihood with the generalized entropy as a regularizer. Such a formulation implies that it can more effectively handle non-separable classification problems, benefiting from the generalized entropy (as confirmed through our experiments). To extend to $K$ classes, we adopt the one-versus-the-rest strategy and then apply a softmax transformation, which gives rise to the posterior probability of the $k$-th class: $p(C_k|\mathbf{x}) = e^{\mathbb{E}[\mathbf{w}_k^T\phi(\mathbf{x})]}/\sum_{j=1}^{K} e^{\mathbb{E}[\mathbf{w}_j^T\phi(\mathbf{x})]}$. The conjugacy introduced by the lower bound function is essential for efficient KMC parameter learning. First, it guarantees a Gaussian form of $q(\mathbf{w})$ and other parameters expressed by the moments of $q(\mathbf{w})$. This allows us to develop an iterative algorithm to efficiently optimize $q(\mathbf{w})$. Second, we can leverage convex duality and the sparse structure to efficiently solve for Lagrangian multipliers.

### 3.1 Parameter Learning in KMC

In order to optimize the KMC objective function in (6) and learn the key model parameters, we first derive the Lagrangian function by introducing dual variables $u_i \geq 0$ and $v$ for each inequality and equality constraints:

$$L(\boldsymbol{\xi}, \boldsymbol{\alpha}, \boldsymbol{\gamma}) = KL(q(\mathbf{w})||p(\mathbf{w}|\boldsymbol{\alpha})) - \mathbb{E}_{q(\mathbf{w})}[\ln h(\mathbf{w}, \boldsymbol{\gamma})]$$

$$+ C\sum_i \xi_i - \sum_i u_i(\mathbb{E}_{q(\mathbf{w})}[y_i(f(\mathbf{w}, \mathbf{x}_i))] + \xi_i) + v(\int q(\mathbf{w})d\mathbf{w} - 1) \quad (7)$$

We start by solving $q(\mathbf{w})$. By setting $\frac{\partial L}{\partial q(\mathbf{w})} = 0$ and recognizing that $q(\mathbf{w})$ takes an exponential quadratic form, we have: $q(\mathbf{w}) = \mathcal{N}(\mathbf{m}_q, S_q)$ with

$$\mathbf{m}_q = S_q[\sum_i (y_i(u_i + \frac{1}{2})\phi(\mathbf{x}_i)], \quad S_q^{-1} = A + 2\sum_i \lambda(\gamma_i)\phi(\mathbf{x}_i)\phi(\mathbf{x}_i)^T, \quad A = \text{diag}(\boldsymbol{\alpha}) \quad (8)$$

We now solve the Lagrangian multipliers. By substituting (8) back to (7), we obtain the dual problem:

$$\max_{\mathbf{u}} - \ln Z(\mathbf{u}), \qquad \text{subject to} \quad \sum_i u_i = C, \quad \forall i, u_i \geq 0 \qquad (9)$$

where $\mathbf{u} = (u_1, ..., u_M)^T$ and $Z(\mathbf{u})$ is the normalization factor that ensures that $q(\mathbf{w})$ integrates to 1. In particular, we have

$$\ln Z(\mathbf{u}) = \ln(2\pi)^{\frac{M}{2}} + \sum_i \ln \sigma(\gamma_i) + \ln |S_q|^{\frac{1}{2}} + \frac{1}{2}\mathbf{z}^T S_q \mathbf{z} + \sum_i (\lambda(\gamma_i)\gamma_i^2 - \frac{1}{2}\gamma_i) \qquad (10)$$

where $\mathbf{z} = \sum_i y_i(u_i + \frac{1}{2})\phi(\mathbf{x}_i)$. By removing terms irrelevant to $\mathbf{u}$, the dual problem is given by

$$\text{Dual (KMC):} \quad \min_{\mathbf{u}} \frac{1}{2}\mathbf{z}^T S_q \mathbf{z} \quad \text{subject to :} \sum_i u_i = C, u_i \geq 0 \qquad (11)$$

The dual problem is essentially a constrained quadratic programming problem of $\mathbf{u}$, which can be solved using a standard QP solver. However, using the ARD prior ensures that the KMC problem has a nice sparse structure, as shown in the following theorem.

**Theorem 1.** *Using an ARD prior, the covariance matrix $S_q$ of variational distribution $q(\mathbf{w})$ has a sparse structure. In particular, for $|\alpha_i| \to \infty$ and $|\alpha_j| \to \infty$, $S_q(i,j) \to 0$ as $S_q(i,j) \propto 1/|\alpha_j|$.*

*Proof.* First, reformulate $S_q^{-1}$ as a matrix form: $A + 2\Phi^T \Lambda \Phi$, where $\Phi = (\phi(\mathbf{x}_1), ..., \phi(\mathbf{x}_M))^T$ and $\Lambda = \text{diag}(\lambda(\gamma_1), ..., \lambda(\gamma_M))$. Given the definition of $\phi(\mathbf{x}_i)$, $\Phi = \Phi^T$. Applying Woodbury identity to $S_q^{-1}$, we have

$$S_q = A^{-1} + A^{-1}\Phi(\Lambda^{-1} + \Phi A^{-1}\Phi)^{-1}\Phi A^{-1} \qquad (12)$$

where $A^{-1} = \text{diag}(\alpha_1^{-1}, ..., \alpha_M^{-1})$ is a diagonal matrix (hence already sparse). So we focus on the second term in (12). In particular,

$$A^{-1}\Phi = \begin{bmatrix} \alpha_1^{-1} & & \\ & \ddots & \\ & & \alpha_M^{-1} \end{bmatrix} \begin{bmatrix} \phi(\mathbf{x}_1^T) \\ \vdots \\ \phi(\mathbf{x}_M^T) \end{bmatrix} = \begin{bmatrix} \alpha_1^{-1}\phi(\mathbf{x}_1^T) \\ \vdots \\ \alpha_M^{-1}\phi(\mathbf{x}_M^T) \end{bmatrix} \qquad (13)$$

Similarly, we have $\Phi A^{-1} = (\alpha_1^{-1}\phi(\mathbf{x}_1), ..., \alpha_M^{-1}\phi(\mathbf{x}_M))$. It can be shown that a significant proportion of $\alpha$'s approach $\infty$ due to the ARD prior [19]. We then apply the Woodbury identity to the term $(\Lambda^{-1} + \Phi A^{-1}\Phi)^{-1}$ in (12) and assume that $|\alpha_i| \to \infty$, it is straightforward to show that $(\Lambda^{-1} + \Phi A^{-1}\Phi)^{-1} \approx A$. Using this fact and (13), we can show $S_q(i,j) \propto 1/|\alpha_j|$ and hence $S_q(i,j) \to 0$ for $|\alpha_j| \to \infty$. $\qquad \square$

Our empirical evaluation over both synthetic and real data shows that a high percentage (e.g., $> 80\%$) of $\alpha$'s are driven to $\infty$ during the optimization process. Therefore, $S_q$ is indeed highly sparse. Thus, the problem can be solved much more efficiently by quadratic solvers boosted by sparse input such as MOSEK [20].

Next, we solve $\boldsymbol{\gamma}$ by set $\frac{\partial L}{\partial \gamma_i} = 0$ and obtain the update rule of $\gamma_i$ as:

$$\gamma_i^2 = \phi(\mathbf{x}_i)^T \mathbb{E}_{q(\mathbf{w})}[\mathbf{w}\mathbf{w}^T]\phi(\mathbf{x}_i) = \phi(\mathbf{x}_i)^T[S_q + \mathbf{m}_q\mathbf{m}_q^T]\phi(\mathbf{x}_i) \qquad (14)$$

The derivation of the update rule of $\boldsymbol{\alpha}$ can benefit from the following result.

**Lemma 2.** *Let $p_1(\mathbf{x}) \sim \mathcal{N}(\mathbf{x}|\mathbf{m}_1, S_1)$ and $p_2(\mathbf{x}) \sim \mathcal{N}(\mathbf{x}|\mathbf{m}_2, S_2)$ then the $KL(p_1||p_2)$ is given by:*

$$KL(p_1||p_2) = \frac{1}{2}\left[\ln \frac{|S_2|}{|S_1|} - M + Tr[S_2^{-1}S_1] + \frac{1}{2}(\mathbf{m}_1 - \mathbf{m}_2)^T S_2^{-1}(\mathbf{m}_1 - \mathbf{m}_2)\right] \qquad (15)$$

Substituting $q(\mathbf{w})$ for $p_1$ and $p(\mathbf{w}|\boldsymbol{\alpha})$ for $p_2$ in (15), we have

$$KL(q(\mathbf{w})||p(\mathbf{w}|\boldsymbol{\alpha})) = \frac{1}{2}(\sum_i \ln \alpha_i^{-1} + \ln |S_q| - M + \text{Tr}[S_p^{-1}S_q] + \mathbf{m}_q^T S_p^{-1}\mathbf{m}_q) \qquad (16)$$

where $S_p^{-1} = A$. Solving for $\frac{\partial L}{\partial \alpha_i} = 0$ while making use of (16), we obtain the update rule for $\alpha_i$:

$$\alpha_i = \frac{1}{S_q(ii) + (\mathbf{m}_q(i))^2} \qquad (17)$$

where $S_q(ii)$ denotes the $i$-th diagonal entry of $S_q$.

## 3.2 KMC-based Multi-class Data Sampling

We develop a novel two-phase KMC-based sampling process to achieve many-class sampling. The proposed KMC model is used for different purposes in each phase: predicting the posterior probabilities of different classes in *initial sampling* and KMC vector discovery for *final sampling*. More specifically, in the initial sampling phase, a pre-trained KMC model using the initial labeled pool is used to make an prediction of all the data samples in the unlabeled pool. The top-$S$ samples will be selected according to their entropy defined over the posterior probabilities of different classes. In essence, these samples *confuse* the current KMC model the most and thus have the greatest potential to improve the model if being labeled. Different from existing AL approaches that directly send these samples for human labeling, the proposed process proceeds by including these samples along with their predicted labels to retrain the KMC model. The goal is to further select data samples identified as KMC vectors, which can contribute to improving the decision boundaries while properly exploring the data space to avoid slow convergence of AL.

We propose a *multi-class sampling function* to measure the overall improvement that a sample can bring to all the classes. In particular, when solving the KMC objective in (6) for the $k$-class, we obtain an optimal $\alpha_i^{(k)}$ for each $\mathbf{x}_i$. Similar to RVM [4], when optimizing each $\alpha_i^{(k)}$, we obtain two quantities $s_i^{(k)}$ and $q_i^{(k)}$ that are referred to as *sparsity* and *quality*, respectively, where sparsity measures the overlap of data sample $\mathbf{x}_i$ with other samples and quality measures $\mathbf{x}_i$'s contribution to reducing the error between the model output and the actual targets. The optimization process will set $\alpha_i^{(k)} \to \infty$ if $q_i^{(k)^2} < s_i^{(k)}$, which makes the corresponding $w_i \to 0$. In this case, $\mathbf{x}_i$ will not contribute to the $k$-class (and is not included as a KMC vector). Otherwise, $\alpha_i^{(k)}$ is set to $s_i^{(k)^2}/(q_i^{(k)^2} - s_i^{(k)})$. Intuitively, if $\mathbf{x}_i$ is not too close to other data samples and effective to reduce the classification error, its corresponding $\alpha_i^{(k)}$ will take a small positive value. By considering all $K$ classes, we use the following function for multi-class sampling:

$$\mathbf{x}^* = \arg\min_i \sum_k \mathbb{E}_{q(\mathbf{w})}[w_i^{(k)}]/\alpha_i^{(k)} \tag{18}$$

In essence, the multi-class sampling function aggregates the impact of $w_i^{(k)}$ that is directly used to compute the posterior probability and the contribution of $\alpha_i^{(k)}$ that gives preference to non-redundant samples that can help reduce the classification error. Finally, by combining the contribution to all classes, it will choose a data sample that can benefit a large number of classes, making it effective and efficient when many classes are involved. The details are summarized in the supplementary materials and the source code is available at [21].

## 4 Experiments

We conduct extensive experiments to evaluate the proposed KMC AL model. We first investigate and verify some important model properties by using synthetic data and through comparison with SVM and RVM, which helps demonstrate the potential of using KMC for AL. We then apply the model to multiple real-world datasets from diverse domains. Comparison with state-of-the-art multi-class AL models will establish the advantage of using KMC in real-world AL applications. For KMC, unless otherwise specified, parameter $C$ is set to $10^{-2}$ and the convergence threshold is set to $10^{-3}$.

### 4.1 Synthetic Data

We draw 500 2-D data samples from a moon-shape distribution and use 70 % of the data for training and 30 % for testing. In Figures 2 and 3, we visualize the learned vectors of the compare models at different noise levels. The results help verify important properties of the proposed KMC model, as described in our theoretical findings. First, the selected KMC vectors sufficiently explore critical areas to cover the entire data distribution while giving adequate attention to the decision boundaries. In contrast, SVM overly focuses on the decision boundaries by using a large number of SVs while RVM under explores the decision areas by using a very small number of RVs and hence suffers from a relatively low model accuracy. This result verifies the desired behavior of KMC vectors that are discovered through optimizing the joint objective function (6). Second, KMC maintains a very high sparsity level at around 90% in both cases. This verifies our theoretical result as stated in Theorem 1. These two important properties clearly establish the potential of using KMC for effective data sampling in AL. Finally, as we add more Gaussian noises to make the data less separable, KMC

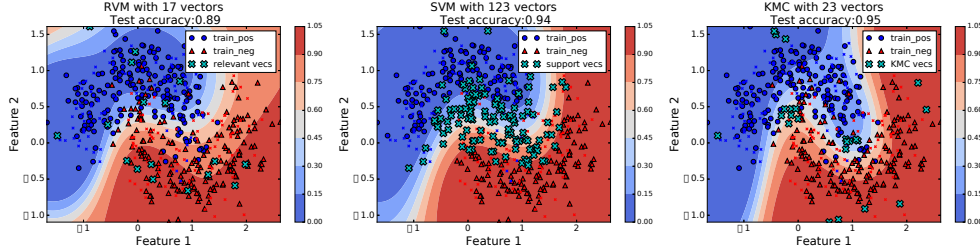

Figure 2: Moon-shape distribution with 30% Gaussian noises

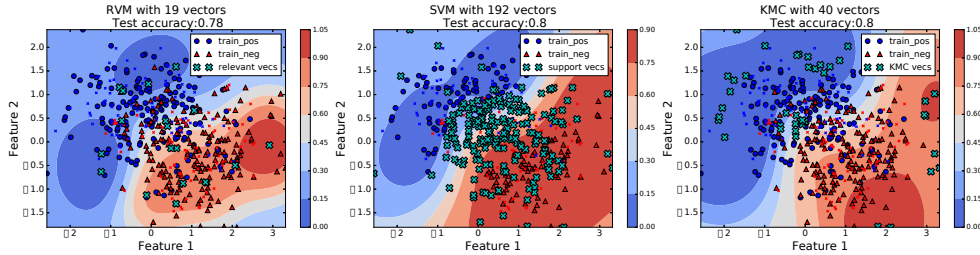

Figure 3: Moon-shape distribution with 60% Gaussian noises

Table 1: Description of Datasets

| Dataset | #Inst | #Attr | #Classes | Class Distr. | Domain |
|---|---|---|---|---|---|
| Yeast | 1484 | 8 | 10 | Skewed | Biology |
| Reuters | 10788 | 5227 | 75 | Skewed | News |
| Penstroke | 1144 | 500 | 26 | Even | Image |
| Derm 1 | 800 | 1391 | 50 | Even | Medical |
| Derm 2 | 868 | 1554 | 30 | Even | Medical |
| Auto-drive | 58509 | 48 | 11 | Even | Automobile |

shows its robustness by maintaining the highest accuracy. Furthermore, the sparsity of KMC remains stable unlike SVM with an exploding number of SVs. This verifies the impact of the generalized entropy regularizer in the KMC objective function (6).

## 4.2 Real Data

We choose 6 datasets from different domains, as summarized in Table 1, to evaluate the proposed KMC based multi-class sampling model.

- **Yeast** uses biological features to predict cellular localization sites of proteins.
- **Reuters** uses the content of Reuters News to conduct text classification.
- **Penstroke** contains handwritten English letters from writers with different writing styles.
- **Derm I&II** contain physicians' verbal narrations of dermatological images with different diseases.
- **Auto-drive** uses different sensor readings to predict failures of a running automobile.

**Experimental setting and comparison methods:**  It is typical for an AL model to start with limited labeled training samples. For datasets with an even class distribution (except for Auto-drive), we randomly select one data sample per class to form the initial training set for AL. For Auto-drive, given its large size, we use 20 labeled samples per class. For unevenly distributed datasets, we randomly sample 1% of the data from each class to form the initial set. For comparison, random sampling (Random) is a commonly used baseline approach. We also include Entropy-based sampling (Entr) that selects the data sample with the maximum entropy of the predicted class distribution. Furthermore, we also compare the KMC AL model with three state-of-the-art multi-class AL models that have been discussed in the related work section, including Best-vs-Second-Best sampling (BvSB) [10], multi-class Probabilistic Active Learning (McPAL) [13], and multi-class convex hull based sampling (MC-CH) [15]. The reported test accuracy is averaged over three runs. Two important parameters of KMC, the large margin coefficient $C$ and the initial sample size $S$, are set to 1 and 40 respectively when compared with other AL models.

**AL performance comparison:**  Figure 4 compares the AL results from KMC and the other four competitive models along with the baseline random sampling. In most cases, the KMC model shows

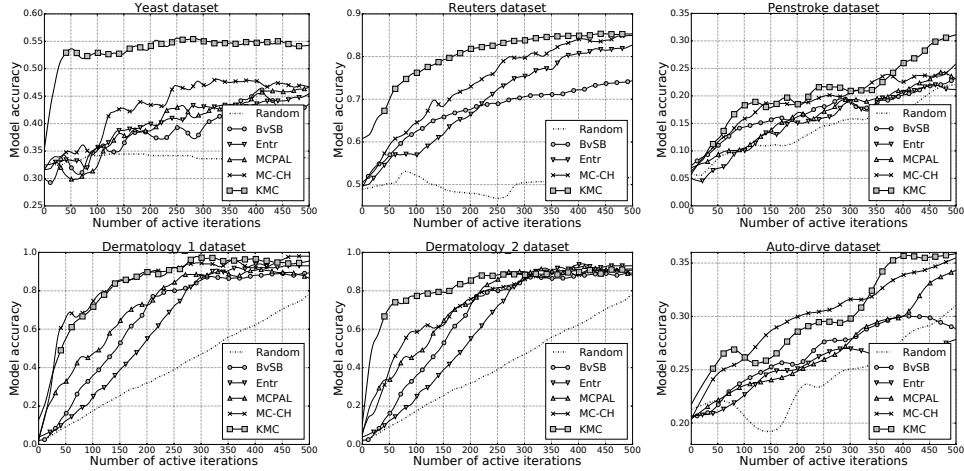

Figure 4: AL Performance Comparison

a fast convergence of AL and better model accuracy. In four datasets, including Yeast, Reuters, Derm I (tied with MC-CH), and Derm II, KMC demonstrates a very clear advantages in the convergence speed of AL. In the other two datasets (i.e., Penstroke and Auto-drive), KMC achieves comparable performance with the best competitive model in the early stage of AL but converges to a high model accuracy in the end. The excellent AL performance of KMC benefits from the ability of the KMC vectors that can effectively explore the entire data distribution while accurately capturing the decision boundaries. The sparse structure of KMC further ensures that only very limited labeled data samples are needed to train highly accurate models. Furthermore, the fast convergence also attributes to effectiveness of the multi-class sampling function (18) that chooses data samples to simultaneously improve multiple classes, which is further verified in later experiments.

**Impact of model parameters:** We study the impact of two parameters, including the large margin coefficient $C$ and the initial sample size $S$ that may affect the model performance. Since a data sample with $\xi_i > 0$ will be wrongly classified, a large $C$ will lead to a variational distribution $q(\mathbf{w})$ with more discriminative strength. In contrast, a smaller $C$ makes the model more tolerant of wrongly classified cases (and also more robust to noises). In practice, KMC performs quite stable over a wide range of $C$ values (i.e., 0.001 to 1) for all the datasets. As for the initial sample size $S$, Figure 5 shows the AL curves of KMC with $S$ set to 5,20, and 40, respectively. It can be seen that at the early stage of AL, a larger $S$ creates an advantage by allowing the model to explore more confusing data samples when the model is not accurate enough to confirm the confusion. For most datasets, such advantage reduces as the model becomes more accurate along the AL process. The only exception is Auto-drive, where a larger initial sample size still shows a good advantage toward the end of the 500 AL iterations. In fact, given the large size of this dataset, after sampling 500 data samples, all the labeled data is only comprised of around 1% of the entire dataset. In addition, this dataset is highly noisy, so the AL model may not fully converge yet by using such limited labeled data, which corresponds to a relative early stage of AL. Therefore, the advantage of using a larger sample size is still significant as in the early stages of other datasets.

**Effectiveness of multi-class sampling:** In this set of experiments, we further verify the effectiveness of the multi-class sampling function by demonstrating that the selected data samples have the potential to benefit multiple classes. Figure 6 visualizes data samples with high sampling scores in the Penstroke dataset from the first 100 AL interactions. For each character, we show the true label along with the most confusing labels (shown in the parenthesis) based on the predictive distribution of KMC. The AL iteration is shown at the bottom. It can be seen that the sampling function prefers samples that is confusing w.r.t. multiple classes. In other words, once the sample is labeled, it has the potential to improve multiple decision boundaries. This exhibits the behavior of more effective exploitation, leading to fast convergence in AL. We also observe that characters in similar appearance tend not to be sampled repeatedly over a large number of AL iterations. Instead, samples from each class are selected in a round-robin manner, which shows the behavior of effective exploration. The good balance between exploitation and exploration explains fast and accurate sampling behavior of the KMC based multi-class sampling function.

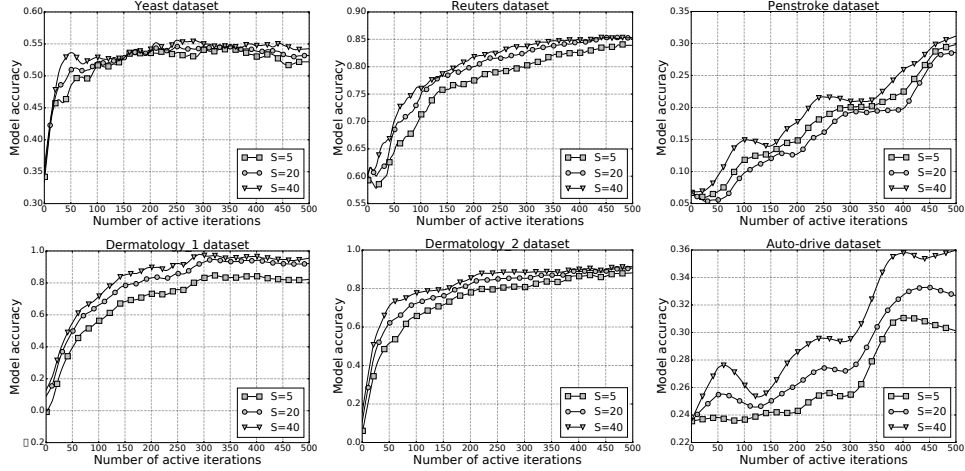

Figure 5: Impact of the Initial Sample Size $S$

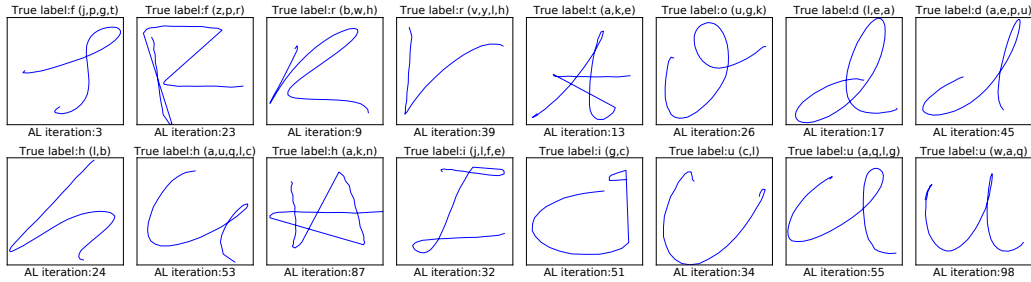

Figure 6: Representative AL Samples that Benefit Multiple Classes

# 5 Conclusion

We propose a novel kernel machine committee that combines Bayesian and discriminative sparse kernel machines for multi-class AL. These two kernel machines with distinct properties are seamlessly unified using the maximum entropy discrimination framework in a principled way that allows the resultant model to choose data samples ideal for AL purpose. The sparse structure of the KMC minimizes the size of the selected data samples for labeling and also ensures efficient parameter learning to support fast model training for real-time AL. A novel multi-class sampling function is designed that combines key model parameters to choose data samples most effective to improve the decision boundaries of multiple classes, leading to faster AL convergence in multi-class problems. Extensive experiments conducted over both synthetic and real data help verify our theoretical results and clearly justify the effectiveness of the proposed KMC based AL model.

## Acknowledgement

This research was supported in part by an NSF IIS award IIS-1814450 and an ONR award N00014-18-1-2875. The views and conclusions contained in this paper are those of the authors and should not be interpreted as representing any funding agency.

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
