[Supplementary Material]

# Supplementary Materials

## KMC-based Multi-class Data Sampling Process

---

**Algorithm 1:** Multi-class Data Sampling using KMC

---

**input** : Initial training set: $(X_l, \mathbf{y}_l)$, unlabeled data pool: $X_u$, kernel function: $k(\cdot, \cdot), S, C$
**output** : Actively learned $K$ KMC models: $\{q^*(\mathbf{w}) \sim \mathcal{N}(\mathbf{m}_q^*, S_q^*), \boldsymbol{\alpha}^*, \boldsymbol{\gamma}^*\}$

1   initialize: Compute $\Phi$ using the given kernel function;
2   **while** $X_u$ **! empty do**
3      **for** *each class $k$* **do**
4          $\boldsymbol{\gamma}^* = \boldsymbol{\gamma}_0, \boldsymbol{\alpha}^* = \boldsymbol{\alpha}_0$;
5          **while !** **converged do**            // Start to train the $k$-th model
6              Evaluate $S_q^*, \mathbf{m}_q^*$ using (8) ;
7              Solve the dual problem (11);
8              **for** $\gamma_i^* \in \boldsymbol{\gamma}^*, \alpha_i^* \in \boldsymbol{\alpha}^*$ **do**
9                  Re-evaluate $S_q^*, \mathbf{m}_q^*$ using (8);
10                 Update $\gamma_i$ using (14);
11                 Re-evaluate $S_q^*, \mathbf{m}_q^*$ using (8);
12                 Update:$\alpha_i^*$ using (17);
13              **end**
14          **end**
15      **end**
16      Predict $p(y|\mathbf{x}_u)$ for $\forall \mathbf{x}_u \in X_u$ ;
17      Compute the entropy: $En(\mathbf{x}_u)$ for $\forall \mathbf{x}_u \in X_u$;
18      Choose top-$S$ data points $X_S$ from $X_u$ according to the entropy measurement.;
19      Retrain KMC with $X_S$ and their predicted labels $\mathbf{y}_S$;
20      Sample $\mathbf{x}^*$ using (18);
21      Label $\mathbf{x}^*$;
22      Move $\mathbf{x}^*$ from $X_u$ to $X_l$;
23 **end**

---

## Passive learning performance comparison

In this set of experiments, we compare the classification performance of KMC with RVM and SVM under the passive learning setting. To demonstrate the behavior of models with different amount training data, we use the same configuration for active learning to initialize passive learning and gradually add new batches of training data. Please note that these new batches are randomly selected instead of relying on a sampling function. This process is repeated either until we observe the convergence of the model performance (Penstroke, Yeast, Auto-drive, and Reuters) or until we run out of training samples (Dermatology 1 and Dermatology 2). As shown in Table 2, the general trend is that with limited training data, RVM and KMC perform better than SVM as SVM may be easily trapped to a locally optimal decision boundary. With sufficient training data, SVM and KMC achieve comparable model performance and both outperform RVM. However, SVM requires a large number of support vectors to fine-tune the decision boundary while KMC uses much less KMC vectors. In summary, in passive learning, KMC can automatically adapt to the size of the training data and provide robust and competitive classification performance in all cases, which mainly benefits from the unified objective function. It also worth to note that KMC is able to clearly outperform the other two models at the presence of 500 new training data samples chosen via active learning. This also shows the effectiveness of the sampling function using the proposed model.

## Sampling method comparison

In this set of experiments, we demonstrate the effectiveness of the proposed sampling function given by (18). This will help justify that the good active learning performance of the proposed model is not only due to the contribution from the KMC model but also benefits from its effectiveness sampling mechanism. More specifically, we report the performance of KMC with best-vs-second best and entropy sampling, respectively. We also include RVM with these two sampling approaches for a

Table 2: Passive Learning Performance Comparison

| Dataset | Model | initial | +100 | +150 | +300 | +500 | +700 | +1000 | +1100 |
|---|---|---|---|---|---|---|---|---|---|
| Yeast | RVM | 0.33 | 0.36 | 0.39 | 0.41 | 0.43 | 0.54 | 0.57 | 0.57 |
| | SVM | 0.32 | 0.35 | 0.35 | 0.36 | 0.39 | 0.59 | 0.6 | 0.61 |
| | KMC | 0.35 | 0.37 | 0.38 | 0.4 | 0.42 | 0.59 | 0.59 | 0.59 |
| Reuters | RVM | 0.53 | 0.52 | 0.53 | 0.53 | 0.54 | 0.54 | 0.58 | 0.58 |
| | SVM | 0.52 | 0.51 | 0.53 | 0.53 | 0.56 | 0.57 | 0.6 | 0.61 |
| | KMC | 0.59 | 0.59 | 0.58 | 0.58 | 0.59 | 0.65 | 0.69 | 0.7 |
| Penstroke | RVM | 0.07 | 0.08 | 0.1 | 0.15 | 0.18 | 0.21 | 0.3 | 0.31 |
| | SVM | 0.06 | 0.12 | 0.13 | 0.17 | 0.21 | 0.23 | 0.32 | 0.34 |
| | KMC | 0.07 | 0.1 | 0.12 | 0.15 | 0.19 | 0.21 | 0.32 | 0.32 |
| Dermatology1 | RVM | 0.12 | 0.2 | 0.25 | 0.46 | 0.78 | 0.95 | N/A | N/A |
| | SVM | 0.06 | 0.17 | 0.24 | 0.48 | 0.8 | 0.96 | N/A | N/A |
| | KMC | 0.16 | 0.22 | 0.26 | 0.45 | 0.78 | 0.95 | N/A | N/A |
| Dermatology2 | RVM | 0.12 | 0.22 | 0.4 | 0.51 | 0.76 | 0.85 | N/A | N/A |
| | SVM | 0.1 | 0.18 | 0.35 | 0.48 | 0.78 | 0.89 | N/A | N/A |
| | KMC | 0.11 | 0.24 | 0.42 | 0.48 | 0.75 | 0.87 | N/A | N/A |
| Auto-drive | RVM | 0.17 | 0.17 | 0.18 | 0.21 | 0.28 | 0.3 | 0.33 | 0.35 |
| | SVM | 0.21 | 0.22 | 0.19 | 0.25 | 0.31 | 0.33 | 0.37 | 0.37 |
| | KMC | 0.19 | 0.19 | 0.19 | 0.22 | 0.28 | 0.3 | 0.34 | 0.35 |

Figure 7: Sampling method comparison

complete comparison. The performance of SVM with the two sampling methods has already been reported in Figure 4 thus omitted here. As shown in Figure 7, the proposed sampling function outperforms the other two commonly used methods. Entropy sampling also provides good performance, which is as expected because entropy is used in the initial sampling phase of our proposed sampling process to identify the top-$S$ candidate samples. The advantage over the entropy based sampling shows the effectiveness of final sampling, which further identifies KMC vectors from the initial candidate set. These KMC vectors effectively contribute to improving the decision boundaries while properly exploring the data space to avoid slow convergence of AL.

Table 3: Efficiency Improvement using the Sparse Structure

|  | Yeast | Reuters | Penstroke | Derm I | Derm II | Auto Drive |
|---|---|---|---|---|---|---|
| Dense | 22s | 111s | 54s | 60s | 35s | 131s |
| Sparse | 12s | 90s | 33s | 45s | 20s | 97s |
| Improvement | 46% | 19% | 38% | 34% | 42% | 26% |

**Computational Efficiency using the Sparse Structure**

We evaluate the computational advantage by leveraging the sparse structure of KMC for parameter learning. To see the benefit of this, we compare the time costs for AL between a dense QP solver, CVXOPT [22], and a sparse solver, MOSEK [20]. Specifically, we measure the computational time for a complete AL sampling step at iterations 50, 100, 150, 200, 300, and 500 and report the average sampling time. In order to make a fair comparison, we use the sparse threshold of $10^{-6}$ to identify zero entities from $S_q$ and the resultant $S_q$ would have sparsity close to $60\%$ for all the datasets. Table 3 shows that AL efficiency can be significantly improved by leveraging the sparse structure of KMC. Notice that the QP-solver needs to be invoked multiple times before KMC converges. Each time, the sparse QP-solver become more efficient due to a better learned, more sparse $\alpha$ while the execution time of the dense QP-solver remains the same. This makes the sparsity of the model more important for fine tuning where more iterations are required to reach a strict convergence condition.