[Reviews · NeurIPS 2019]

Reviewer 1



The algorithm is based on an optimization problem that is well motivated and can be tackled using QP solver. This seems to clearly outperform several baselines. The main drawback of this paper is lack of theoretical guarantees of the active learning procedure. Typically, one would want to prove generalization and label complexity bounds.

Reviewer 2



The idea of combining RVM and SVM is interesting, especially in the context of active learning, since it encourages selecting points both near the decision boundary and exploring the input space well. The paper is well written. The experiments seem to demonstrate advantages over baselines. The usage of the word "generative" in this paper might be inappropriate. I think RVM is a probabilistic method, but not generative since it doesn't model P(x | y). See wikipedia for several common definitions of "generative" and "discriminative" models. In Figure 4, why in some of the plots (e.g., for Yeast and Reuters datasets), the learning curves started from different accuracy? Not all active learning methods use the KMC learning algorithm? If it's not, then this raises the question of whether the advantage of the proposed method is due to the AL policy or the learning algorithm. If it is the same learning algorithm, then all learning curves should start from the same accuracy. Are the experiments averaged over multiple repeats or just a single repeat? It's not mentioned the experiments are repeated, so I'm concerned about the statistical significance of the results. How large is S in the experiments on the real datasets (Figure 4)? Table captions should be on top of the tables. [update]: from the author's rebuttal, it seems the proposed KMC algorithm appear to outperform SVM or RVM. As different active learning policies are coupled with different learning algorithms, given the reported better performance of the proposed KMC on passive learning setting, it becomes unclear what contributed to the better performance of active learning: is it the policy or the learning algorithm? The new learning algorithm itself could still be a nice contribution. But as an active learning paper, I think the authors should more analysis to make this clear.

Reviewer 3



Originality: The combination of sampling in areas of 'greater interest' while adjusting to the underlying distribution appears in many active learning works, but the objective in (1) is novel and approaching both in a unified framework is challenging. The lower bounding of the optimization problem is also new Quality: The experimental results are very thorough and show the improvement of the proposed method over random sampling as well as several other baselines. And the exploration of effect of tuning parameters and initial sample size is excellent. However the theoretical contributions appear incomplete. The significant theoretical contribution is the (mislabelled) Theorem 2, and both the statement and proof of this is extremely informal. This should be made much more formal. And since the sparsity structure is integral to the computational efficiency of the method this is a useful and important Theorem. Clarity: The paper makes a clear line from the motivation to the setup. However the derivation of (1) is quite difficult to follow, even though the expression itself is intuitive. And the flow from expression (1) to the final expression at (8) is well structured. It is not entirely clear to the reader that the simulations in Figure 2 and Figure 3 are showing what the authors claim they show. Here the excess exploration by KMC appears unnecessary, and an example where the SVM excessively samples an area with very low density might be more useful. The real experiments do a good job of showing the efficacy of the proposed method. The paper has a few typos (see Theorem 2) but overall they did not detract from the readability. Significance: Optimizing for both informativeness and relevance in active learning is a difficult task, and unified approaches like this are a good avenue for approaching the problem. However in order to maximize impact it would be good for the approach to provide some sort of guarantees about the quality of their solution, even just in simple situations. Edit: If the full proof sketched in the response is added to the paper I am happy to increase my recommendation to an accept, especially since in practice the authors reported that the sparsity rate was around 80% (though could they please confirm this was across all the data sets, especially the real world ones).

[Author Response · NeurIPS 2019]

**Reply to Reviewer 1**

We thank the reviewer for the constructive suggestion and recommendation for acceptance. We appreciate your positive evaluation on the novelty of the proposed unified AL model along with the sampling function and the convincing experimental results. We also agree with the reviewer about the importance of generalization guarantees and label complexity bounds, which will be an exciting direction that we plan to pursue in our future work. We would like to point out that the proposed model introduces a much more complex (regularized) loss function in order to integrate two different types of sparse kernel machines for more effective sampling and uses a specially designed sampling function for multi-class problems. Addressing these challenges in the theoretical analysis could help further extend the state of the art on the theory of active learning.

**Reply to Reviewer 2**

We thank the reviewer for the thoughtful feedback and recommendation for acceptance. We appreciate your positive evaluation on the novelty of the optimization algorithm for the new learning objective and the AL policy.

*Q1: RVM is a probabilistic, not a generative model.* We agree that RVM does not model $p(x|y)$ and our initial intent is to leverage RVM's capability to model the conditional distribution of the response $y$ given the input $x$. We will make this clear as suggested by the reviewer.

*Q2: The learning curves started from different accuracy and not all active learning methods use KMC.* The different starting accuracy is caused by different learning models. In fact, for many AL methods, the sampling rules are designed for specific learning algorithms. In our proposed approach, the sampling rule given in eq. (13) is developed along with the KMC model as it uses model-specific information for sampling. Due to this coupling, the selected data sample can help improve the given model to the largest extent. The same rationale also applies to several other competitive models in our experiments. For example, MC-CH is built upon SVM as it uses the convex hull of support vectors for sampling. Similarly, McPAL requires its own learning model and BvSB is typically used with an SVM model.

*Q3: Should report average or median results in experiments.* The reported test accuracy is averaged over three runs. We will make this clear in the revised paper.

*Q4: How large is S in the experiments on the real datasets.* S is set to 40, which will be made clear.

*Q5: Compare RVM, SVM, and KMC in passive learning setting.* Following the reviewer's suggestion, we have compared these models in passive learning using the 6 real-world datasets. The general trend is that with limited training data, RVM and KMC perform better than SVM as SVM may be easily trapped to a local optimal decision boundary. With sufficient training data, SVM and KMC achieve comparable model performance and both outperform RVM. However, SVM requires a large number of support vectors to fine-tune the decision boundary while KMC uses much less KMC vectors. In summary, in passive learning, KMC can automatically adapt to the size of the training data and provide robust and competitive classification performance in all cases, which mainly benefits from the unified objective function.

**Reply to Reviewer 3**

We thank the reviewer for the constructive feedback. We appreciate your positive evaluation on the novelty of the objective function/lower bounding of the optimization problem and thorough experimental results.

*Q1: Theorem 2 should be made much more formal.* We will provide a more formal statement and proof of the Theorem (and also fix the label) as suggested by the reviewer. In particular, the Theorem can be more formally specified as " Using an ARD prior, the covariance matrix $S_q$ of variational distribution $q(\mathbf{w})$ has a sparse structure. In particular, for $|\alpha_i| \to \infty$ and $|\alpha_j| \to \infty$, $S_q(i,j) \to 0$ as $S_q(i,j) \propto 1/|\alpha_j|$ (similarly $S_q(j,i) \to 0$ as $S_q(j,i) \propto 1/|\alpha_i|$)". We will provide a reference to Faul, A. C., & Tipping, M. E. (2002), which proved that some $\alpha$'s approach $\infty$ to ensure the sparsity of RVM (also verified in our experiments). We will also provide more details for the proof to make it clear that $S_q(i,j) \propto 1/|\alpha_j|$ for $|\alpha_i|, |\alpha_j| \to \infty$. A key step added to the current proof is to apply the Woodbury identity to the term $(\Lambda^{-1} + \Phi A^{-1}\Phi)^{-1}$ in eq.(17) and by using the fact that $|\alpha_i| \to \infty$, we can show $(\Lambda^{-1} + \Phi A^{-1}\Phi)^{-1} \approx A$. Using this fact and eq. (18), we can show $S_q(i,j) \propto 1/|\alpha_j|$ and hence $S_q(i,j) \to 0$ for $|\alpha_j| \to \infty$.

*Q2: Derivation of (1) is difficult to follow.* The key insight of objective function (1) is to combine a likelihood term that well captures the data distribution with a large margin constraint to simultaneously ensure good discriminative power of the model. The regularizer is added to help ensure model sparsity.

*Q3: It is not entirely clear that simulations in Figures 2 and 3 show the authors' claim.* The main purpose of these figures is to show that KMC sufficiently explores critical areas of the data distribution while giving adequate attention to the decision boundaries by using limited KMC vectors. While KMC uses slightly more vectors than RVM, it is much sparser than SVM. The middle chart of Figure 3 shows excessive support vectors are assigned close to the decision boundary (including some low density areas) while KMC only assigns a few vectors there as shown in the right chart.

[Meta-Review · NeurIPS 2019]

The paper proposed a novel algorithm for active learning in the multi class setting. The authors present a theoretical guarantee regarding the sparseness of the model as well as empirical evaluation across 6 datasets and comparing with 5 baseline methods. All reviewers tend for vote for acceptance, but do point out several areas of improvement and the authors provide feedback for. I strongly expect the final version of the paper to include the changes that address: - Formal statement and proof outline for Theorem 2. - Include the comparison of RVM, SVM and KMC method in the passive learning setting (mentioned in the author feedback), in order to help distinguish the benefit of the novel model alone, in addition to the combination of model and active sampling. - BvSB could also be used with the KMC/RVM method. Including those results would significantly increase the value of the study.